# Evaluating and Improving Neonatal Gentamicin Pharmacokinetic Models Using Aggregated Routine Clinical Care Data

**DOI:** 10.3390/pharmaceutics14102089

**Published:** 2022-09-30

**Authors:** Dominic M. H. Tong, Jasmine H. Hughes, Ron J. Keizer

**Affiliations:** InsightRX, San Francisco, CA 94104, USA

**Keywords:** gentamicin, TDM, MIPD, neonates, continuous learning, pharmacokinetics, population pharmacokinetic modeling

## Abstract

Model-informed precision dosing (MIPD) can aid dose decision-making for drugs such as gentamicin that have high inter-individual variability, a narrow therapeutic window, and a high risk of exposure-related adverse events. However, MIPD in neonates is challenging due to their dynamic development and maturation and by the need to minimize blood sampling due to low blood volume. Here, we investigate the ability of six published neonatal gentamicin population pharmacokinetic models to predict gentamicin concentrations in routine therapeutic drug monitoring from nine sites in the United State (*n* = 475 patients). We find that four out of six models predicted with acceptable levels of error and bias for clinical use. These models included known important covariates for gentamicin PK, showed little bias in prediction residuals over covariate ranges, and were developed on patient populations with similar covariate distributions as the one assessed here. These four models were refit using the published parameters as informative Bayesian priors or without priors in a continuous learning process. We find that refit models generally reduce error and bias on a held-out validation data set, but that informative prior use is not uniformly advantageous. Our work informs clinicians implementing MIPD of gentamicin in neonates, as well as pharmacometricians developing or improving PK models for use in MIPD.

## 1. Introduction

Gentamicin is an aminoglycoside antibiotic used as part of a first-line therapy for sepsis in neonates [1,2]. Due to its narrow therapeutic index, high inter-individual variability in pharmacokinetics, and substantial drug exposure-related risks of ototoxicity and nephrotoxicity [3,4,5,6,7], therapeutic drug monitoring (TDM) is recommended to individualize dosing and obtain optimal exposure [8]. TDM-guided dosing in neonates is, however, complicated due to their dynamic development and maturation processes, especially of the kidneys, where gentamicin is primarily eliminated, and by the limited blood volume of neonates, which constrains the number of blood samples that can be drawn.

Using model-informed precision dosing (MIPD) in conjunction with TDM holds particular promise for neonates in that some of this dynamic maturation process and interindividual variability of this vulnerable patient population can be accounted for using model covariates. However, MIPD relies on using an adequately predictive population pharmacokinetic (popPK) model to describe the patient population [9]. Multiple popPK models for gentamicin in neonates have been published [10,11,12,13,14,15]. Crcek et al. [16] have recently compared model structures and assessed dosing recommendations from these publications. However, to our knowledge, there has not yet been an external meta-analysis of model predictiveness in a routine clinical care data set, nor any attempts to improve existing model predictiveness.

One method of improving model predictiveness for MIPD is continuous learning, in which an initial model is used in MIPD and then refit as additional data become available [17,18,19]. Previous work has shown a 3–13% reduction in error for vancomycin models using data from only 200 patients [17], and significant improvements in target attainment for busulfan in a prospective dosing study [18]. Models can be refit using the published model parameters as an informative prior, or can be fit without a prior, i.e., refit entirely based on the new collected data. It is not yet clear whether the use of a prior improves model performance more than a refit on the new data alone. 

Here, we investigate the performance of published popPK models for gentamicin in neonates in routine clinical care data from nine sites in the United States. We refit these models with and without an informative prior from the published models in a continuous learning process and evaluate the performance of these refit models against the published models.

## 2. Materials and Methods

### 2.1. Study Population

Routine clinical care data for 788 patients treated at nine sites in the United States were entered into the InsightRX Nova platform. These nine sites are acute care hospitals or academic teaching hospitals in California, the Midwest, and the East Coast of the United States. De-identified data were included in this analysis if the patient was between 0 and 12 weeks old and had at least 1 recorded serum gentamicin concentration. A total of 313 patients were removed for the following reasons: missing information on serum gentamicin concentrations, imputed or missing gestational age, gentamicin concentration measurements during or within 15 min of end of infusion, unrealistic drops in gentamicin concentration (of 2 mg/L or more within 30 min, indicating a data entry error), or assumed inaccurate dosing records (e.g., multiple large doses within 10 min). Current weight, serum creatinine, gestational age, and postnatal age were collected for each patient.

### 2.2. Population Pharmacokinetic Model Literature Review

A literature review was conducted to identify population pharmacokinetic models describing gentamicin PK in neonates for clinical use (Figure 1). Models were included in this analysis if they were: (1) developed on a population closely aligned with the data set described here [9], i.e., majority neonatal population; (2) developed on a non-specialized population (i.e., not hypothermic or septic); (3) used routinely collected covariates; (4) developed in NONMEM; (5) not superseded by a newer model fit; and (6) developed after the year 2000. The last criteria derives from our experience in implementing and evaluating over 200 popPK models from the literature, from the estimation methods being more approximate before 2000 [20], and from the lack of modern consensus on what constitutes proper model development and diagnostics in models developed before 2000 [21,22,23].

Clinical practice for gentamicin is to sample at the peak and at the trough, commonly targeting peaks between 8 and 12 mg/L and troughs less than 1 mg/L for optimal therapy [8]. Dose adjustments are made using samples paired or grouped in this way. To best capture clinical practice, model predictions were evaluated iteratively: the first group of gentamicin levels were predicted using population estimates; subsequent levels were grouped similarly and predicted prospectively, using all preceding samples to estimate the individual’s PK parameters, and then using these parameters to predict the grouped sample (see Figure 2, purple box). A group of gentamicin levels was defined as (a) a peak and a trough during the same dosing interval, or (b) a trough and peak on either side of a dose, or (c) a single sample that was not collected close in time with another sample.

### 2.3. Population Pharmacokinetic Model Refitting and Evaluation

Patients were randomly assigned into the training (70% of data) or the testing (30% of data) data sets (see Figure 2) and data distributions were checked to confirm that gentamicin sample types and patient covariates were roughly equal between sets. The popPK models with lowest predictive error on the full data set (Bijleveld, Fuchs, Garcia, and Wang models) were refit using NONMEM on the training data, with or without an informative prior. Model and covariate structures were not modified from the published models, except for removing the covariate effect of dopamine from the Fuchs model, as dopamine co-administration was not available in the data. Any parameter that could not be estimated due to numerical instability or that converged to zero was fixed to the published value instead. The refit models with and without informative prior were compared to the published model on error and bias with the testing data set.

An informative prior was implemented using the NONMEM $PRIOR NWPRI subroutine, with the variance of the parameter priors ($*THETAPV*) set according to the standard error (*RSE*) and the parameter *THETA* in the published model:(1)$$THETAPV={\left(\frac{RSE}{100}\cdot THETA\right)}^{2}$$

The degrees of freedom of the interindividual variability prior ($OMEGAPD) was set to 50 for all models, a somewhat informative prior [24].

### 2.4. Statistics and Error Metrics

The predictive performance of each model was evaluated using root mean square error (*RMSE*) and mean percent error (*MPE*), using the ith prediction ($pred_{i}$) and observation ($obs_{i}$), as follows:(2)$$RMSE=\sqrt{\frac{\sum_{i=1}^{N}{\left(pred_{i}-obs_{i}\right)}^{2}}{N}}$$
(3)$$MPE=\frac{1}{N}\overset{N}{\underset{i=1}{\sum }}\frac{pred_{i}-obs_{i}}{obs_{i}}\times 100\%$$

Uncertainty in these error metrics was assessed by bootstrapping samples and computing the error metric across these samples. Statistical significance was determined by the overlap of the 5th to 95th percentiles of bootstrapped samples of performance parameters. Analysis of data files generated by NONMEM (version 7.4.4, ICON Development Solutions, Ellicott City, MD, USA) and PsN (Perl-speaks-NONMEM) version 5.2.6 [25] was performed in R version 4.1.0.

## 3. Results

### 3.1. Patients and Data Collection

The resulting data set comprised 475 patients (193 female) treated at 9 sites, as summarized in Table 1. The median (range) gentamicin dose was 12.4 (1.6–24.7) mg or 4.2 (0.96–5.8) mg/kg (see Appendix A). There were 304 peak samples (measured between 0.5 and 2.5 h after the start of infusion), 466 trough samples (measured within 2 h of the next dose) and 72 samples that were neither a peak nor a trough (“other”). A median (range) of 2 (1–8) serum gentamicin concentration measurements were taken per patient. 

### 3.2. Evaluation of the Literature Models

Covariate distributions of collected data were compared to those of the literature models (Figure 3, Table 2). The De Cock and Wang models were developed on patient populations including older patients, leading to higher weights and postnatal ages. The other model development populations generally matched the covariate distributions in our data, although the median gestational age was younger for all models except the Wang model.

The six selected popPK models for gentamicin in neonates were evaluated for error and bias aggregated by sample timing (peaks, troughs, and other) and by population-level (a priori) versus individualized (a posteriori) PK parameters (Figure 4). Peaks showed higher error than troughs across all models, which is expected since peaks are higher in magnitude than troughs. The published models were also evaluated for accuracy, defined here as whether the model predicted troughs to within 0.5 mg/L or peaks to within 2 mg/L (Figure 5). These thresholds were considered clinically relevant allowable error margins. 

The De Cock and Germovsek models produced significantly lower precision and higher bias than the other models considered. The De Cock model, which includes only weight as a predictor of gentamicin PK (Table 2), showed a strong correlation between prediction error and age metrics (i.e., postnatal, postmenstrual, and gestational age), suggesting a mis-specified covariate model (Appendix A). The higher error observed for these models could arise from differences between the model validation data set described here and the model development data sets. While both models were developed in data sets containing neonatal patients, the De Cock model development population contained children up to 15 years of age. The Fuchs model showed low error in predicting troughs but a high degree of error in a posteriori peak predictions. Eliminating the rather high covariance term in interindividual variability between clearance and volume of distribution (87% correlation) improved prediction performance for both a posteriori peak and trough prediction, resulting in the Fuchs, no covariance (“Fuchs NC”) model. The Bijleveld, Fuchs NC, Garcia, and Wang models, which were developed on patient populations comparable to the one described in this paper (Table 1) and showed a good balance of high accuracy (Figure 5), low error, and low bias (Figure 4) for model re-estimation, were selected for further consideration.

### 3.3. Model Re-Estimation

The four models with lowest error were refit on the training data set, with the published model parameters used as informative priors (“prior”) or without a prior (“refit”). Re-estimated model parameters are shown in Figure 6 (see also Appendix A). Informative prior constrains the model parameter to be nearer to the published value, except in cases where the signal is moved to another parameter. For the Bijleveld model, the informative prior constrained the parameters closer to the published values, except for IIV in the central volume of distribution. In the Fuchs model, the effect of postnatal age on clearance is higher in our data than in their published data set, but using an informative prior to constrain the parameter for the effect of postnatal age on clearance meant individual variability moved to other parameters (IIV on clearance and additive error) instead.

### 3.4. Re-Estimated Model Evaluation

The refit models were evaluated against the published models on the held-out validation data set. The refit models showed equal or better predictive performance compared to their published counterparts (Figure 7). The use of an informative prior generally reduced error and bias in the Fuchs and Bijleveld models, whose model development population postnatal ages better matched those in this study. In the Wang model, a refit without an informative prior reduced error and bias, though this comes at the increase in covariance of inter-individual variability of clearance and volume, a sign of overfitting as noted with the Fuchs model.

## 4. Discussion

While model-informed precision dosing (MIPD) holds promise for gentamicin in neonates, a meta-analysis on how well previously published models perform in routine clinical care in this patient population has been lacking. Here, we evaluated the error and bias of six published popPK models in a large data set of 475 patients aged 0.36–81.9 days treated at nine sites in the United States. The four models with the highest precision and lowest bias were then refit to evaluate if predictive performance could be improved further. Finally, the impact of using the published model as an informative prior in this continuous learning process was assessed.

We evaluated model predictive performance using only data available to the clinician at the time of dosing, to ensure our analysis reflects actual clinical performance [26]. There are a limited number of a posteriori gentamicin samples collected, leading to higher variability in error and bias estimates for a posteriori samples. However, population (a priori) estimates are critical for initial dosing and rapid target attainment [22], with most gentamicin treatment courses shorter than 48 h [10]. Model-to-model differences in prediction error also tend to attenuate for a posteriori samples [17]. Therefore, the results from this large and diverse set of neonates across nine sites in the United States should generalize well across similar populations and to clinical practice.

Complex models do not necessarily translate to improved predictive performance in real-world clinical data on new patients. The Germovsek model used many samples across many patients to fit a three-compartment model, exploring the physiological PK of gentamicin, while the Wang model considers neonatal development in some detail, with a maturation function and mean creatinine based on postnatal age. These more complex models may describe the drug concentration curve over the whole dosing interval better; however, in practice, these models perform no better in predicting peaks and troughs than the Bijleveld model, a simpler, more empiric two-compartment model with two covariates trained on a data set 68–97% smaller. This suggests popPK model development for clinical use is less dependent on the number of compartments or parameters of a model than it is about collecting relevant covariates and sufficient samples per patient in a diverse patient population.

Selecting a model for clinical use depends not only on patient demographic data and biomarkers, but also on drug coadministrations such as indomethacin and ibuprofen in neonates, on biological characteristics such as degree of development (for which gestational and postmenstrual ages are a proxy), and on other conditions such as therapeutic hypothermia. Here, we find that the best models for a general population of neonates (that may contain specialized patients) are the Bijleveld, Fuchs NC, or Wang models, as these models have the lowest error and bias across all predictions. The Garcia model performs well a priori but has significantly higher error a posteriori than the other three models. However, the best model for a population may not be the best model for an individual patient. For a patient that is not well-described by a model, we recommend the clinician evaluate why the model is fitting poorly. Are the patient covariates outside of the range of the model development population? Are similar patients in terms of gestational age at birth or critically ill status included in the model development population? Could co-medications be causing an interaction effect? The solution may be to select another model that better reflects patient covariates or subpopulation such as premature neonates, or to use clinical judgment in adjusting from the model predictions. We also recommend sampling drug concentrations earlier and more frequently if feasible for these patients, to improve pharmacokinetic parameter estimates and predict future drug concentrations more accurately.

Previous work has shown that vancomycin, a drug with a rich popPK literature, can benefit from continuous learning on populations as small as 200 patients [17]. We demonstrate here that for gentamicin, another drug where covariates impacting pharmacokinetics are well-known, continuous learning on 332 patients improves model prediction accuracy. This is especially important for population predictions, since most gentamicin treatment courses are shorter than 48 h and therefore do not require TDM-based dose individualization. For less well-understood drugs or drugs with higher interindividual variability, a continuous learning process may require more patient data, or that a new model with an alternate structure or covariate model be developed. 

We refit models with and without an informative prior from the published model [24], using the $PRIOR subroutine in NONMEM. Previous studies have used $PRIOR to boost sparsely sampled but large data sets [27], to incorporate previous PK studies on a different population [28], or to increase available information using prior studies [29], but do not compare the reported model with the same model fitted without the prior. Using an informative prior during model fitting can be thought of as a way to prevent model parameter estimates from adapting too strongly to potentially noisy data by anchoring them to known values. Future work should explore how the differences between prior and current population affect informative prior use, how training data size impacts this decision, and how optimizing the informative prior weighting can improve model predictions. 

Continuous learning improved predictive performance of popPK models for gentamicin in neonates, but the implementation of these models into clinical practice will require ensuring models continue to accurately fit the patient population by monitoring for model drift and overfitting. Covariates can vary between sites but also within sites over time, leading to a phenomenon commonly known as model drift. Model drift may arise from changes in the population itself, such as demographic shifts or from changes in clinical practice, such as changes in dosing, co-medications, sampling assays, or other protocols. 

Overfitting, where the model fits the training population too well, leads to models that do not generalize to future patients. The high covariance between clearance and volume of distribution in interindividual variability in the Fuchs model may be an example of overfitting, which lead to poor predictive performance in a new population, especially for peak samples. Monitoring data from incoming patient populations and comparing these data to the model development population, as well as changes in error and bias of predictions, are critical in detecting model drift and overfitting. We demonstrated here the first step in a continuous learning process to improve model predictions for gentamicin in the vulnerable neonatal population, potentially reducing the need for additional blood samples for therapeutic drug monitoring of gentamicin. Subsequent implementation of continuous learning models for the clinic must focus on the reliability, generalizability, and trustworthiness of these models for clinicians at the point of care.

## Figures and Tables

**Figure 1 pharmaceutics-14-02089-f001:**
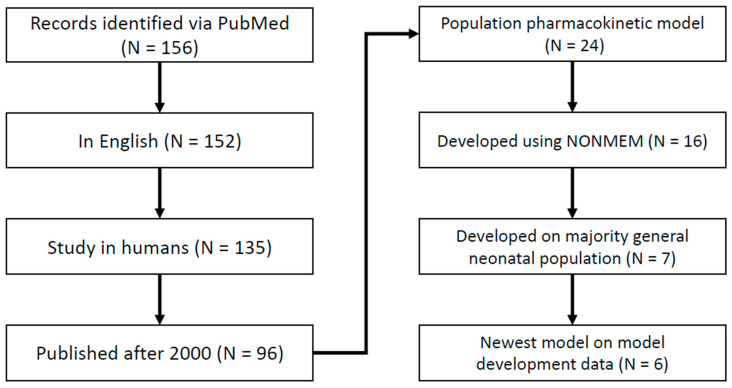
Diagram depicting the literature search for population pharmacokinetic models for gentamicin in neonates in this study.

**Figure 2 pharmaceutics-14-02089-f002:**
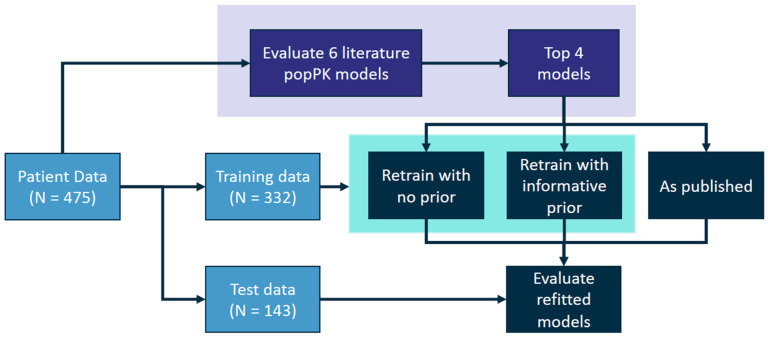
Diagram depicting the experimental design of this study. Six population pharmacokinetic models from the literature were evaluated (purple box). The three models with lowest error and bias were retrained with and without priors (teal box) and evaluated on a held-out test data set.

**Figure 3 pharmaceutics-14-02089-f003:**
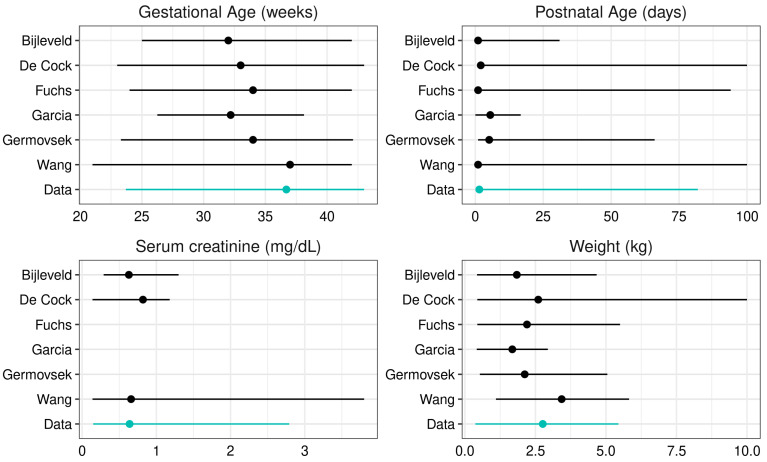
Model development population covariate distributions. Dots (lines) represent median (range), except for Garcia covariates, which are given as mean (two times standard deviation). Postnatal age is cropped to 100 days and weight cropped to 10 kg for clarity.

**Figure 4 pharmaceutics-14-02089-f004:**
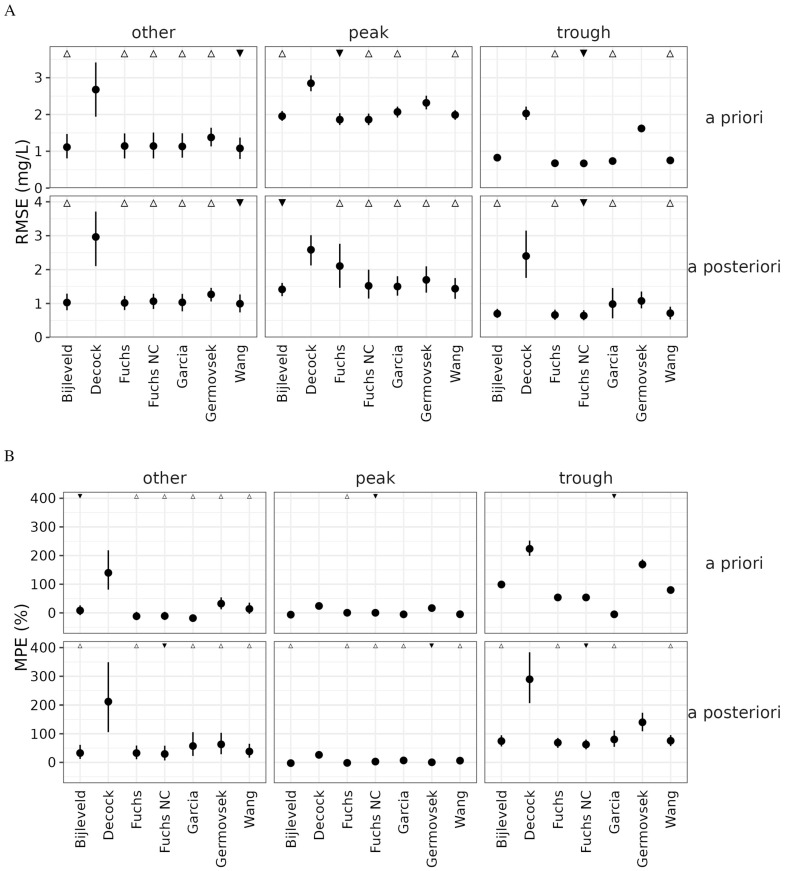
Prediction error of gentamicin population pharmacokinetic models for neonates, assessed by (**A**) root mean square error (RMSE) and (**B**) mean percent error (MPE). Closed circles (vertical bars) represent the median (5th to 95th percentile) of 1000 bootstraps. Statistical significance was determined by presence of overlapping confidence intervals: ▼ represents the model with lowest median RMSE in that category; △ represents models not statistically different from the best model. Fuchs NC: Fuchs, no covariance.

**Figure 5 pharmaceutics-14-02089-f005:**
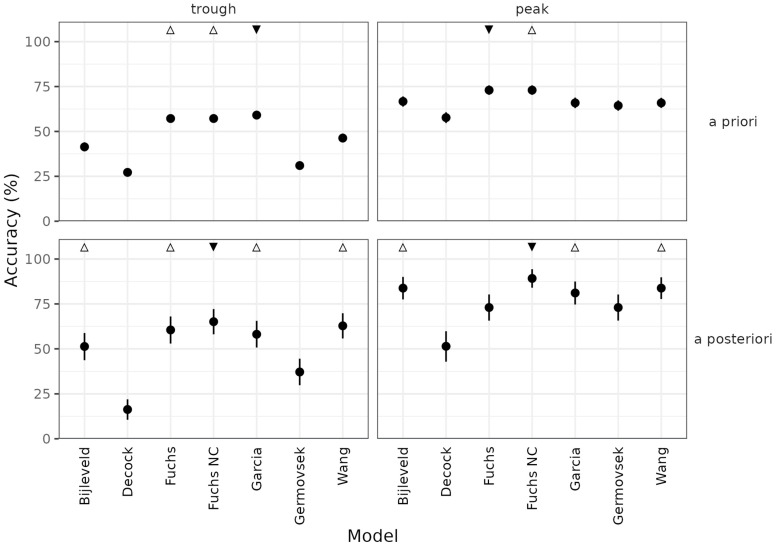
Accuracy of published gentamicin models in predicting target attainment for troughs and peaks, stratified by population (“a priori”) or individualized (“a posteriori”) predictions. Trough predictions are accurate if the prediction is within 0.5 mg/L of the actual value. Peak predictions are accurate if the prediction is within 2 mg/L of the actual value. Statistical significance was determined by presence of overlapping confidence intervals: ▼ represents the model with highest median accuracy in that category; △ represents models not statistically different from the best model. Fuchs NC: Fuchs no covariance.

**Figure 6 pharmaceutics-14-02089-f006:**
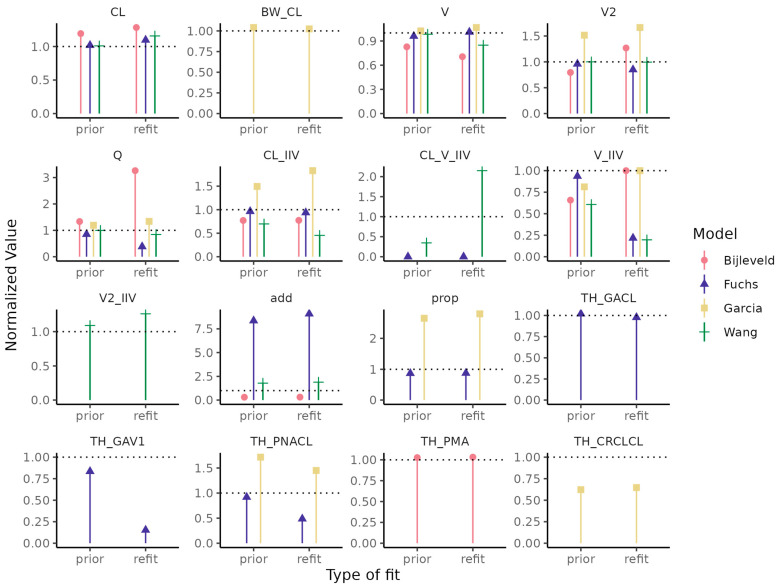
Parameter estimates, normalized to the published value, for the refit with informative prior (“prior”) and refit without prior (“refit”) versions of each of the Wang, Bijleveld, Fuchs, and Garcia popPK models. CL: Clearance; BW_CL: effect of body weight on CL; V: volume of distribution of central compartment; V2: volume of distribution of peripheral compartment; Q: intercompartmental clearance; IIV: interindividual variability; add: additive error; prop: proportional error; TH_PMA: effect of PMA on CL; TH_GACL: effect of GA on CL; TH_GAV1: effect of GA on V1; TH_PNACL: effect of PNA on CL, TH_CRCLCL: effect of creatinine clearance (CRCL) on CL.

**Figure 7 pharmaceutics-14-02089-f007:**
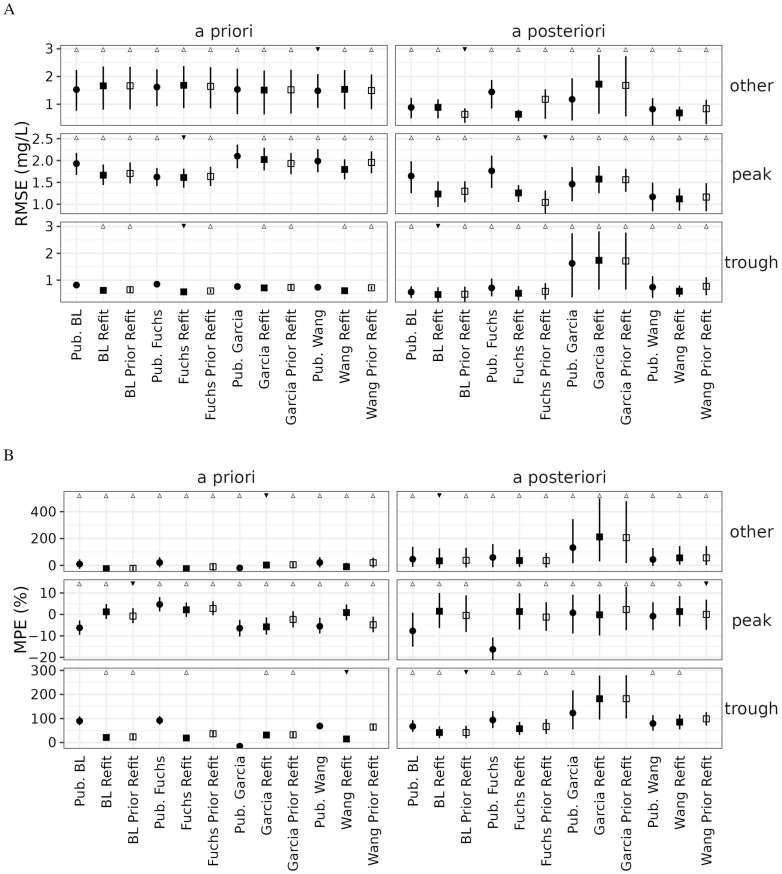
(**A**) Root mean square error (RMSE) and (**B**) mean percent error (MPE) on testing data for published models (circles), fully refit models (filled squares), and models refit with $PRIOR in NONMEM (hollow squares). Closed circles (vertical bars) represent the median (5th to 95th percentile) of 1000 bootstraps. Statistical significance was determined by presence of overlapping confidence intervals: ▼ represents the model with lowest median RMSE or lowest absolute median MPE in that category; △ represents models not statistically different from the best model. BL: Bijleveld.

**Table 1 pharmaceutics-14-02089-t001:** Patient characteristics of the overall data set, the training data set for model re-fitting, and the validation data set for re-fitted model evaluation. Values are median (range).

Characteristic	Overall	Training Data	Validation Data
Number of Patients	475	332	143
Number of Sites	9	9	9
Number of TDMs	842 (304 peaks, 466 troughs, 72 other)	590 (210 peaks, 328 troughs, 52 other)	252 (94 peaks, 138 troughs, 20 other)
Age (days)	1.46 (0.36–81.9)	1.46 (0.36–81.9)	1.46 (0.36–41.1)
Gestational age (weeks)	36.7 (23.7–43)	36.6 (23.9–43)	37 (23.7–41.4)
Serum creatinine (mg/dL)	0.64 (0.15–2.79)	0.64 (0.15–2.79)	0.63 (0.19–2.05)
Weight (kg)	2.76 (0.37–5.44)	2.67 (0.37–4.94)	2.86 (0.49–5.44)

**Table 2 pharmaceutics-14-02089-t002:** The literature models describing neonatal gentamicin pharmacokinetics selected for evaluation and collected data covariates. Current study data covariates are also described. Values are given as counts (for patients, samples), median (range) or mean (standard deviation). PNA postnatal age; GA gestational age; PMA postmenstrual age; WT current weight; HT height; CR serum creatinine; CRCL creatinine clearance; NG not given.

Model	Pts	Samples	PNA (d)	GA (wk)	WT (kg)	CR (mg/dL)	Model Covariates
Bijleveld [13]	65	136	1 (0–31)	32 (25–42)	1.84 (0.43–4.67)	0.63 (0.29–1.3)	WT, PMA
De Cock [14]	717	1705	2 (1–5475)	(23–43)	2.6 (0.44–80)	0.82 (0.14–1.18)	WT
Fuchs [10]	1518	3039	1 (0–94)	34 (24–42)	2.2 (0.44–5.5)	NG	WT, GA, PNA, dopamine coadmin
Garcia [15]	200	417	5.49 (5.41)	32.19 (2.97)	1.68 (0.63)	NG	WT, PNA, CRCL
Germovsek [11]	205	1325	5.1 (1–66)	34 (23.3–42.1)	2.12 (0.53–5.05)	NG	WT, GA, PNA, CR
Wang [12]	2357	6359	1 (1–6924)	37 (21–42)	3.43 (1.1–5.83)	0.66 (0.14–3.8)	WT, HT, CR, PNA, PMA
Current Study Data	475	842	1.46 (0.36–81.9)	36.7 (23.7–43)	2.76 (0.37–5.44)	0.64 (0.15–2.79)	-

## Data Availability

Scripts to recreate continuous learning models have been provided as Appendix A.

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
