# Peer review of "Evaluating and Improving Neonatal Gentamicin Pharmacokinetic Models Using Aggregated Routine Clinical Care Data"

_pharmaceutics, 2022, doi:10.3390/pharmaceutics14102089_

Round 1
Reviewer 1 Report
Dear authors
The article describes the external evaluation of five published neonatal gentamicin PKpop models in order to inform clinicians implementing MIPD. The authors shows how to select model by using standard approach. Among selected models with acceptable levels of bias, precision and accuracy for clinical use, they show that refitting models reduce error and bias with their own data. This is an interesting approach, however, there are some major/minor issues or missing references that should be drastically reconsidered on the whole article.
Major issues:
1. Section: Materials and Methods à Study population
Ethics: regarding the use of data patient, do you have an institutional review board to allow you to use these data in compliance with the latest recommendations for the protection of individuals?
2. Section: Materials and Methods à population pharmacokinetic model evaluation
Please rename this sub-section Literature search as it is corresponding to selection process.
The rationale for model selection is described very succinctly. Literature search is not exhaustive and is not well described knowing that there are at least 10 models matching your selection criteria (I only talk about models with a NONMEM approach, your selection in blue):
-Wang et al. 2019 (1)
-Bijleveld et al. 2017 (2)
-Germovsek et al. 2016 (3)
-Medellin-Garibay et al. 2015 (4)
- Fuchs et al. 2014 (5)
- De Cock et al. 2014 (6)
- Lopez et al. 2010 (7)
- Nielsen et al. 2009 (8)
- Garcia 2006 (9)
- Jensen et al. 1992 (10)
- Bryson et al. 1988 (11)
If the models I quote in addition to yours do not match your selection criteria then you must argue. Please detail your search in a sub-section: Literature search with mesh terms and search engines such as Pubmed, Scopus, Web of Science... In Results section, Please begin with a subsection named: Literature search including A flowchart of studies included for evaluation detailing the rationale and inclusion/exclusion criteria and then including your table 1. It is required for the reader's clarity. Please put the paragraph (line 84-93) in your sub-section Population pharmacokinetic model refitting and evaluation.
3. Section: Materials and Methods à Population pharmacokinetic model refitting and evaluation.
Line 125: there is a big mistake concerning your formula:
You forgot to divide by n. I hope this is an oversight. Please check all your results.
(the right formula)
4. Section: Results à Evaluation of literature models
In evaluation of predictive performance, it would be necessary to detail:
- Goodness-of-fit by showing the results of the comparison of population prediction and individual predictions to the observations for each model and yours, graphically and numerically (R², MPE, RMSE for pop and ind.).
- Residuals error by showing graphically QQplot, NPDE and pcVPC for the evaluated models and numerically make a table summarizing mean, variance, and results of normality and symmetry tests of the NDPE distribution for the evaluated models.
5. Section: Conclusion
What is missing from your discussion is a summary of what are the best models to use for someone who wants to Bayesian foerecasting in a clinical department and what they should do to minimize prediction error.
Minor issues:
line 85: or this paragraph you have made mistakes, probably a typo: “between 8 and 12 mg/dL and troughs less than 1 mg/dL”. Please correct: between 8 and 12 mg/L and troughs less than 1 mg/L
Line 97 : Five population pharmacokinetics not Six
Figure 1: Patient Data, N = 475 and when you split the data 14 patients are missing (N=461= Training D 323 + Test D 138). How do you explain that?
References
1. Wang H, Sherwin C, Gobburu JVS, Ivaturi V. Population Pharmacokinetic Modeling of Gentamicin in Pediatrics. J Clin Pharmacol. déc 2019;59(12):1584‑96.
2. Bijleveld YA, van den Heuvel ME, Hodiamont CJ, Mathôt RAA, de Haan TR. Population Pharmacokinetics and Dosing Considerations for Gentamicin in Newborns with Suspected or Proven Sepsis Caused by Gram-Negative Bacteria. Antimicrob Agents Chemother. janv 2017;61(1):e01304-16.
3. Germovsek E, Kent A, Metsvaht T, Lutsar I, Klein N, Turner MA, et al. Development and Evaluation of a Gentamicin Pharmacokinetic Model That Facilitates Opportunistic Gentamicin Therapeutic Drug Monitoring in Neonates and Infants. Antimicrob Agents Chemother. août 2016;60(8):4869‑77.
4. Medellín-Garibay SE, Rueda-Naharro A, Peña-Cabia S, García B, Romano-Moreno S, Barcia E. Population pharmacokinetics of gentamicin and dosing optimization for infants. Antimicrob Agents Chemother. janv 2015;59(1):482‑9.
5. Fuchs A, Guidi M, Giannoni E, Werner D, Buclin T, Widmer N, et al. Population pharmacokinetic study of gentamicin in a large cohort of premature and term neonates: Population pharmacokinetics of gentamicin in newborns. Br J Clin Pharmacol. nov 2014;78(5):1090‑101.
6. De Cock RFW, Allegaert K, Brussee JM, Sherwin CMT, Mulla H, de Hoog M, et al. Simultaneous Pharmacokinetic Modeling of Gentamicin, Tobramycin and Vancomycin Clearance from Neonates to Adults: Towards a Semi-physiological Function for Maturation in Glomerular Filtration. Pharm Res. oct 2014;31(10):2643‑54.
7. Lopez SA, Mulla H, Durward A, Tibby SM. Extended-interval gentamicin: Population pharmacokinetics in pediatric critical illness: Pediatr Crit Care Med. mars 2010;11(2):267‑74.
8. Nielsen EI, Sandström M, Honoré PH, Ewald U, Friberg LE. Developmental Pharmacokinetics of Gentamicin in Preterm and Term Neonates: Population Modelling of a Prospective Study. Clin Pharmacokinet. juin 2009;48(4):253‑63.
9. Garcia B. Population pharmacokinetics of gentamicin in premature newborns. J Antimicrob Chemother. 30 mai 2006;58(2):372‑9.
10. Jensen PD, Edgren BE, Brundage RC. Population pharmacokinetics of gentamicin in neonates using a nonlinear, mixed-effects model. Pharmacotherapy. 1992;12(3):178‑82.
11. Bryson SM, McGovern EM, Kelman AW, Whiting B, Thomson AH, Way S. Population Pharmacokinetics of Gentamicin in Neonates. Dev Pharmacol Ther. 1988;11(3):173‑9.

Reviewer 2 Report
This manuscript compared five published models to predict gentamicin peak and trough concentrations in neonates. The author selected 3 models that outperform the rest, and ran refit of all the 3 models. However, it is not obvious that the refit has improved the performance significantly. Here are some comments:
1. Line 68, about the dataset composition, please clarify what is unrealistic drops in gentamicin concentration, what is criteria to define it and any scientific evidence to make this criteria?
2. Line 68, for assumed inaccurate dosing records, please clarify what are considered as assumed inaccurate dosing records.
3. Line 1001, what approach was used to randomize data into training and testing set? And what was performed to avoid bias in assigning those data?
4. In Fig 3, the De Cock can be easily excluded based on the poor prediction performance. But for other approaches, like Germovsek, it is difficult to tell. Did the author have any specific criteria?
5. Line 183, the improvement of Fuchs NC from Fuchs is not very obvious. And why this approach can be improved but not the others?
6. Similarly, in Fig 4 (line 185), how do the author draw the conclusion of good balance of high accuracy (Figure 4), low error, and low bias?
6. Line 222-223, it is hard to say that the refit model for Wang is better than published, do the author have some evaluation criteria to demonstrate it?
7. In Fig 6, did the author use Fuchs NC or Fuchs method? Why?
Reviewer 3 Report
I have read this paper with great interest, and a background as clinician-neonatologist, and clinical pharmacologist, so involved in the use of MIPD, but less in the ‘back offices’ of these tools.
Based on this background, my main concern rather relates to the extrapolation to different units, with different practices and case mix, like eg asphyxia cases with therapeutic hypothermia, the use of either indomethacin or ibuprofen in these populations, micro-preemies, weight as current weight, or birth weight, or creatinine assays (beyond the range considered, in eg figure 2. How relevant in this case mix to the performance exercise ? At present you have used 70/30%, but I assume that this a reflection of the ‘average population’, and does not necessary apply to specific subcategories of neonatal patients ?
I suggest that the authors are also more explicit in their text on either birth weight, or current weight ?
The approach you suggest in the introduction (continuous learning) is therefore also rather ‘unit’ specific, or do you have sufficient confidence on this aspect ?
Was the literature search ‘systematic’, or rather pragmatic ? what platform(s) was/were used ?
Reviewer 4 Report
Review comments on pharmaceutics-1854755: Evaluating and improving neonatal gentamicin pharmacokinetic models using aggregated routine clinical care data
The manuscript described the investigation of neonatal gentamicin popPK models to predict gentamicin concentrations in neonates. The authors found three models with acceptable levels of error and bias for clinical use. This study is of high novelty and contribution to the PK field. However, there are several issues in this manuscript that require clarifications or modifications prior to publication. Please consider the following comments.
1. Study population: the authors should include details of clinical data employed in this study, such as data source and name of 9 sites.
2. Table 1: It seems to have no data given as mean (standard deviation). Please correct the table title.
3. Figure 1: the legend mentions “Six population pharmacokinetic models from literature were evaluated”, which is inconsistent with other places.
4. Figure 1 shows training data (N = 323) and test data (N = 138). The sum is only 461, not 475 as stated.
5. Data in Table 1 were repeated in Figure 2.
6. Figure 3: please clarify the L-P model.
7. Line 241: it should be 0.36-81.9 days
8. Citation style should be modified following the journal guidelines.
Round 2
Reviewer 1 Report
The manuscript has been revised following my recommendations.
Reviewer 2 Report
The author has addressed the comments raised.
Reviewer 4 Report
The manuscript was appropriately revised. There was only a problem with the citation style. In-text citation should be "neonates [1,2]" with a space, not "neonates[1,2]".